# A Comprehensive Review on Printed Electronics: A Technology Drift towards a Sustainable Future

**DOI:** 10.3390/nano12234251

**Published:** 2022-11-29

**Authors:** Sridhar Chandrasekaran, Arunkumar Jayakumar, Rajkumar Velu

**Affiliations:** 1Center for System Design, Department of Electronics and Communication Engineering, Chennai Institute of Technology, Kundrathur, Chennai 600069, India; 2Green Vehicle Technology Research Centre, Department of Automobile Engineering, SRM-Institute of Science and Technology, Kattankulathur 603203, India; 3Additive Manufacturing Research Laboratory (AMRL), Indian Institute of Technology Jammu, Jammu 181221, Jammu & Kashmir, India

**Keywords:** printed electronics, sustainable manufacturing, future of manufacturing, industry 5.0 3d-printing, semiconductor processing

## Abstract

Printable electronics is emerging as one of the fast-growing engineering fields with a higher degree of customization and reliability. Ironically, sustainable printing technology is essential because of the minimal waste to the environment. To move forward, we need to harness the fabrication technology with the potential to support traditional process. In this review, we have systematically discussed in detail the various manufacturing materials and processing technologies. The selection criteria for the assessment are conducted systematically on the manuscript published in the last 10 years (2012–2022) in peer-reviewed journals. We have discussed the various kinds of printable ink which are used for fabrication based on nanoparticles, nanosheets, nanowires, molecular formulation, and resin. The printing methods and technologies used for printing for each technology are also reviewed in detail. Despite the major development in printing technology some critical challenges needed to be addressed and critically assessed. One such challenge is the coffee ring effect, the possible methods to reduce the effect on modulating the ink environmental condition are also indicated. Finally, a summary of printable electronics for various applications across the diverse industrial manufacturing sector is presented.

## 1. Introduction

In the present world, semiconductor device technology is dominating the growth of the economy in the electronic manufacturing industry. As the semiconductor industry is one of the biggest industries with a wider range of products around the globe with a net worth of 646 billion dollar market by 2022 [1]. A typical electronic transistor has a semiconductor as channel material, conducting metal as the electrodes, and insulator as the gate dielectric. For complete integrated circuit process technology, a silicon wafer needs to use multiple-stage processes governed by complex and expensive machinery. Moreover, the semiconductor device fabrication process with metal etching and lift-off process introduces huge metallic waste into our environment. Traditional electronics require complex machinery involving metal etching and lift-off processes which leads to a rise in toxic metal and acid post-fabrication [2]. A leap toward sustainability with proper waste management can be achieved by incorporating manufacturing technology with reduced waste.

Furthermore, we need a promising technology that drives the manufacturing sector with the industrial 4.0 revolutions. As a consequence, there is a need for a smart manufacturing technology with minimal electronic waste through a sustainable process and AM can be one such potential option. Industries are expected to utilize AM techniques to transform the two-dimensional traditional manufacturing process into a three-dimensional process. Another important advantage of using AM is the ability to fabricate different kinds of materials such as metals [3], insulators [4], semiconductors [5], ceramics [6], and polymers [7]. In addition, AM is expected to play a significant role in semiconductor chip packaging [8] and metal interconnects [3]. As reported by Callen Votzke et al., 3D printed, metal-based interconnects show stable conductivity for 350 cycles of strain test [9]. The surface energy of the printing ink should be lesser than that of the substrate for even printing with clear resolution. The surface energy of the generally used printable electronics substrates is observed to be ~36 mNm^−1^ on Si/SiO_2_ & glass [10] and the PET-based flexible substrate has a surface energy of ~52 mNm^−1^ [11]. The surface energy has a direct impact on the evaporation and distribution of the printed ink. Controlling the surface energy of the printed ink by using a lower surface energy base such as deionized water/alcohol will minimize the coffee ring effect and accelerate the evaporation process after printing [12].

Figure 1a shows the number of publications in the web of science database by using printable electronics as the search keyword and its evident that the database shows a gradual rising trend in publications. To further analyze the ink involved in printing, we conducted another ink-based search using the keywords such as 2D materials ink, metal nanoparticle ink, and metal oxide nanoparticle ink. Once again, the results show a positive trend for 2D-materials ink and metal nanoparticle ink while metal oxide nanoparticle and other ink variants are at the early stage of research.

In this review, we holistically access the diverse manufacturing technologies to replace the traditional semiconductor process and their cross-functional applications published in the past 10 years. In addition, we discuss the various ink technologies based on metal nanoparticles, metal oxide nanoparticles, 2D materials, molecular, and resin-based ink. Moreover, we discussed in detail the various printing technologies such as inkjet printing, aerosol jet printing, extrusion printing, electrohydrodynamic printing, and light-based printing.

## 2. Materials and Process: Assessment

The attributes of printable electronics require high-performance ink and apparently should fulfill the desired role based on the applications such as sensors, radio frequency devices, flexible displays, and energy storage devices [13]. Generally, conducting ink is highly suitable for the fabrication of electrodes and it covers a broad range of applications in electronic devices, energy storage devices, solar panels, metamaterials, and antennas [14]. A semiconducting material is usually preferred as an active material for transistors and tactile sensors [15,16]. Insulating ink is preferred for the transistor gate terminal and dielectric materials for supercapacitors or other energy storage devices [17,18]. Printable ink can be broadly classified as metallic nanoparticle ink, metal oxide nanoparticle ink, 2D-material ink, molecular ink, and resin-based ink.

### 2.1. Metallic Nanoparticle Ink

Metallic nanoparticle inks are composed of metal nanoparticles, organic solvents, and stabilizing agents. On heating the coated metallic nanoparticle ink, the insulating liquids are evaporated to form a nanoparticle-based high metallic layer which is highly desirable for conducting metal lines, and electrodes [19]. Metallic nanoparticle inks are advantageous because of their superior conductivity, magnetic properties, stability to oxidation, and low-cost alternative. The metallic nanoparticle ink can be customized for tuning the electrical or magnetic properties of the ink based on the target application. For a good electrode, a highly conducting ink is generally preferred which could be obtained by choosing a conducting metal nanoparticle such as Au [20], Ag [21], Cu [22], Al [23], Co [24], Ni [25], Pt [26], Zn [27], Pd [28], and Sn [29]. Ironically, the metal nanoparticle ink is not suitable for high-temperature devices as they oxidize on exposure to air. However, nanoparticle inks based on metal oxides are desirable for electronics applications and are discussed in detail in the next section.

### 2.2. Metal Oxide Nanoparticle Ink

Nanoparticle ink is based on metal oxide nanoparticles which are highly stable and developed from the metal oxide nanoparticle based on copper oxide nanoparticles [30], indium tin oxide [31], zinc oxide [32], and iron oxide nanoparticles [33]. In general, metal oxide nanoparticle inks are low cost than pure metallic nanoparticle ink and simple to manufacture without any concern about oxidation [19]. Copper oxide nanoparticle ink is generally used for highly conductive metal lines and to achieve high conductivity the surface oxide is removed by treating with the reducer (reduction process) [34,35]. The nanoparticle ink is protected with a coating to the sintered pattern to avoid post-deposition oxidation. Indium tin oxide-based conductive ink is used for transparent electronics applications with optical transmittance of 98% [36]. The ITO-based conductive ink requires high-temperature sintering at 300 °C to achieve desired electrical conductivity [36]. ZnO is also known very well as an eco-friendly and sustainable material with zero impact on the environment. Zinc oxide is a semiconducting nanoparticle ink that acts as an alternative to ITO-based ink because of its low cost and low-temperature process [37]. Zinc oxide nanoparticles exhibit unique properties suitable for optoelectronics, photovoltaics, electronics, and various sensing applications [38]. IZO ink exhibits semiconducting nature making it desirable as a thin film transistor [39]. Iron oxide-based metallic ink is highly magnetic ink with good electrical properties [40]. The iron oxide-based metallic inks are suitable for the fabrication of printable patch antennas, inductors, and other radio frequency devices [41].

### 2.3. 2D-Material Ink

Recently 2D-material has been gaining rapid attention because of their low production cost and mass production alternative using solution-based processing technology. The growth of 2D materials on wafers is believed to be a challenging task and requires a high chemical vapor deposition instrument with a higher thermal budget. The printable ink using 2D materials is based on nanoflakes such as graphene [42], MoS_2_ [43], WS_2_ [44], MoSe_2_ [45], black phosphorous [12], and h-BN [46]. Ink formulated based on h-BN is a good gate material for transistors because of its wide bandgap nature [47]. 2D-materials ink based on graphene can be used as the conducting as well as semiconducting materials which is an alternative for transistors channel and conductive electrodes for flexible electronic devices [48,49]. A study reported by Tian Carey et al. used porous h-BN and graphene-based ink for the fabrication of the transistor on a textile-based substrate. 2D materials such as MoS_2_ and BP are indirect bandgap materials that are highly suitable for optoelectronics applications such as light emitters and photodetectors [50,51,52,53,54]. Moreover, the 2D materials exhibit nonlinear properties under optical light which further makes them suitable for nonlinear optical switches [55].

### 2.4. Molecular Ink

The metal and organic composition are the fundamental building blocks for the development of molecular ink and are generally based on the MOD process to produce high-quality conductive thin films on thermal sintering. The MOD-based ink is advantageous over other ink because of being particle-free, low-cost, and easily adaptable to various printing techniques. Bhavana Deore et al. [56] proposed a screen printable Cu-based molecular ink with higher printing resolution, mechanical durability, and a highly conductive path printed on the sheets of the flexible substrate (Kapton and PET). The Cu MOD ink will be a low-cost alternative to nanoparticle-based conductive ink and it is highly suitable for electronics applications such as printable antennas, bonded LEDs, and transistor’s printable electrodes with excellent electrode stability on exposure to air ambiance for 6 months.

In a study, Arnold Jason Kell et al. [57] developed a silver-based conductive molecular ink for printing flexible electronics with a sheet resistance of <10 mΩ/sq/mil and producing highly metallic conduction under extreme bending and a flexible environment. The authors also tested the feasibility of the metal traces, in developing a capacitor, inductor, and capacitor-based 3rd-order Chebyshev filter with a cut-off frequency of 1 GHz. Moreover, the conducting ink’s flexible nature makes them compatible with inkjet, aerosol jet printing, and screen-printing methods which further opens the array of opportunities in the development of electronic devices of next-generation printable electronics.

### 2.5. UV-Curable Ink

UV-curable ink is attracting attention in textile industries due to its mechanical strength and durability. In general, UV-curable conductive ink is prepared by mixing the UV-curable chemicals with the metallic nanoparticle to achieve evenly distributed metallic nanoparticles and a UV-curable base. UV-curable inks are required to be cured at low temperatures and have shorter curing times for utilization in a wide range of applications. In a study, Hong et al. [58] proposed a screen-printed UV-curable conductive ink based on silver nanoflakes and polymer-based reins. The UV-curable conductive ink is patterned as a confirmable antenna operating as ultra-high frequency RFID tags on the textile-based substrate. The textile-based stretchable conductive fabric will be revolutionizing the future of wearable electronic devices as the recent advancement on UV-curable ink based on silver@polypyrrole with superior structural integrity, toughness, and high conductivity on cotton fabric substrate [59]. In another study, the authors investigated the silver-based nanoparticle ink by screen printing technology on various nylon fabrics with superior electrical conductivity which is highly desirable for wearable electronic applications [60].

## 3. Printing Technologies

The printing of complex structures requires designing tools to design the patterns with the exact dimensional data of the device. In general, the design of the printing pattern is developed by using commercial tools such as computer-aided design, and computer-aided manufacturing tools to build the 3D structure of any complex structure which are essential for feeding the printing machine. The printing of conductive layers can be obtained by using various printing techniques such as (a) inkjet printing, (b) aerosol printing, (c) filamentary printing, (d) electrohydrodynamic printing, and (e) UV-light curing-based printing as shown in Figure 2. Table 1 shows the comparison of various nanoparticle ink using printing technology.

### 3.1. Inkjet Printing

The inject printing operation is based on the ejection of microdroplets from the printing head and the nozzle undergoes pressure changes by the formation of the microbubbles and collapse as shown in (Figure 2a). The printing ink should be non-viscous for reducing the shear force during ejection from the nozzle and the viscosity of the ink should be less than 100 mPa-s for optimal printing [72]. Conversely, the inject printer based on a piezoelectric actuator controls the ink flow by contraction and expansion of the piezoelectric actuator. The inkjet printing technique is widely used to print a variety of materials such as metal nanoparticles, conducting graphene, and metal oxide nanoparticles. For a good inkjet printing system, the conducting ink of higher solubility is highly preferred to minimize the printing head clogging and improve the overall printing pattern. In a study, the 2D material based on h-BN and graphene layers is grown using an inkjet printing system as shown in Figure 3.

Figure 3a shows the process of the heterojunction transistor begins with the bottom gate metal PEDOT: PSS used on textile substrate and silver for the polymer substrate). The graphene and h-BN are used as the channel material for the heterojunction transistors printed on the textile. Finally, the source and drain electrodes are patterned by printing using PEDOT: PSS for textile substrate and silver for polymer substrate as shown in (Figure 3a). Figure 3b shows the FE-SEM cross-sectional image of the device showing the presence of PEDOT: PSS, h-BN, and graphene layers on the Textile substrate. The proposed heterojunction FET is also fabricated on the polymer substrate with an inverter and OR logic circuit. Figure 3c shows the optical microscopic image of the p-type and n-type transistors connected to form the inverter logic gate on the flexible polymer substrate. The electrical response of the inverter on applying a square waveform is depicted in Figure 3d. On increasing the device input voltage, the resistance response shows the functionality of the p and n operation as marked in the red line of Figure 3e. OR-logic functionality of the transistors by using two additional resistors and their transient response is depicted in Figure 3f,d.

### 3.2. Aerosol Jet Printing

The aerosol jet printing works based on the atomization of ink by ultrasonication method as shown in (Figure 2b). The ultrasonic energy on the ink generates aerosol with highly desirable characteristics for jetting the aerosol. The carrier gas drives the aerosol from the ink chamber to the deposition head. Aerosol is printed onto the sample surface by driving the aerosol with sheath gas. The focusing ratio is the ratio of the sheath gas flow rate to the carrier gas flow rate as shown in Equation (1). The focusing ratio determines the quality of the metal line printed as reported by Ankit Mahajan et al. [67]. In this study, a high-resolution silver metal line was printed on a flexible polyimide substrate with a line resolution of 40 µm using aerosol jet printing as shown in Figure 4a–i. The focusing ratio directly influences the width and thickness of the silver metal line as depicted in Figure 4j. On increasing the focusing ratio and stage speed, the width of the metal line is decreased. In addition, by increasing the focusing ratio and decreasing the stage speed the thickness of the printed line increases. Furthermore, the printing efficiency of aerosol jet printing can be improved by controlling the focusing ratio of the aerosol jet printer tuned for modulating the line width of the printed line [67]. The aerosol jet printing technology is a promising alternative technology for the fabrication of metal interconnects via semiconductor ICs [73].
(1)$$\mathrm{Focusing}\;\mathrm{ratio}=\frac{Sheath\;gas\;flow\;rate}{carrier\;gas\;flow\;rate}$$

In another study reported by Kihyon Hong et al. [64], they used aerosol jet printing technology to print the P-type and N-type transistors based on ZnO as the channel material as depicted in Figure 4k–o. This shows the promising nature of aerosol jet printing technology in semiconductor manufacturing industries. The aerosol jet printing is applicable for micro-scaled devices but not suitable for cutting-edge technology nodes due to the technology scaling limitations.

### 3.3. Extrusion-Based Printing

The functionality of extrusion or filamentary-based printing is similar to the FDM system which is based on the high-temperature melting of the fuse on controlled motions across the (x, y, z) plane. It is a layer-by-layer extrusion-based printing technology that is used for the accurate fabrication of any 3-dimensional layered structures [74]. The extrusion-based printed electronics are highly suitable for printing a complex 3D microstructure with superior electrical properties [75,76,77]. Figure 5 depicts the graphene-based flexible conducting line deposited on the surface of the glove for sensing strain, pressure, and EMG sensors. 3D-printed Cu-based filament has been extensively used to build printed electrical interconnects, metallization, and circuits [78]. In addition, the authors also studied the multiple-layered metal interconnects and their electrical performance across three embedded interconnect metal layers. The electrical conductivity of the interconnects is further tested by PIC controller-based printed circuit board to further evidence the capability of digital data transmission across the printed metal lines. A study reported by Leland Weiss and Tyler Sonsalla [79], focused on the fabrication of perovskite solar cells using an FDM process and the fabricated perovskite solar cells have a cell size of 25 mm with 200 µm as the MAPbI_3_-PCL thickness. Perovskite materials tend to obtain improved conductivity and transparency at elevated temperatures due to increased electron-hole pair generation. Fabrication of perovskite solar cells further realizes the possibility of utilizing commercial applications because of the simple and efficient process technology offered by FDM. Thus, the FDM technology used in the fabrication of solar cells will open new pathways for the rapid fabrication of highly efficient and simple process technology for solar cell manufacturing.

### 3.4. Electrohydrodynamic Printing

Electrohydrodynamic printing is based on the redox reaction of metallic ink and the sample surface under the influence of an electric field [80,81,82]. Figure 6a depicts the in situ deposition of metal Mo acting as the anode which is immersed in the acetonitrile solvent and the metal ions M^z+^ generated inside the printing nozzle. The generated metal ions are deposited on the substrate as reduced metal ions and the ions transfer is controlled by the DC voltage of 80–150 V to drive the redox printing. Alain Reiser et al. demonstrated the two-channel nozzle composed of Cu and Ag ion species and positive voltage is applied to any one of the wire electrodes then the Cu^+^ or Ag^+^ ions deposited on the surface as shown in Figure 6b,c. Similarly, when both channels are activated by positive voltage then alloys of Cu–Ag are formed on the surface. Figure 6d shows the EDX elemental map of the Cu and Ag-based material grown on the substrate which further evidence the controllability of metal deposition on varying the actuating voltage from channel to channel. Figure 6e shows the array of 50 × 50 nanopillars made of Cu which are spaced 500 nm apart along the x and y axis and this implies the x-y axis motion controllability of the printer in nanometer scale. Figure 6f depicts the high-resolution Cu metal lines with various line spacing of 1 µm, 500 nm, and 250 nm. The authors showcased the dimensional controllability with high accuracy by electrohydrodynamic printing which is demonstrated by printing the highly dimensionally scalable structures of 85 and 170 nm rod width as shown in Figure 6g,h. It shows the formation of nanorods, metallic lines, elevated nanorods, and sinusoidal wave-like patterns are demonstrated in Figure 6i–k.

Jaehyun Bae et al. [83] investigated the effect of the surface tension of printable ink on the jetting flow of redox printing. The authors used water, EG, DMSO, DMF, acetone, ethanol, and IPA of varying surface tensions from 72.8 (dyne/cm) to 20.9 (dyne/cm) as shown in Figure 7. The jetting flow for water is semi-circular meniscus due to the higher surface tension for the water of 72.8 dyne/cm. Meniscus dimension changes from semi-circular to conical on decreasing the surface tension as observed for DMSO with 42.9 dyne/cm. The jetting flow with the conical-shaped meniscus when the surface tension is less than 42.9 dyne/cm which is observed for surface tension as low as 20.9 dyne/cm. Henceforth, the meniscus exhibits directional jet flow for DMSO, DMF, acetone, ethanol, and IPA. Similarly, the authors controlled the thickness of the droplet by increasing the applied voltage to 4 kV.

### 3.5. Light-Based Printing

UV curing has been a popular technique used in additive manufacturing technique for building complex 3D structures. UV curing techniques have received increased interest recently due to their fast curing, high resolution, energy efficiency, less space, and solventless process. UV curing ink works based on the photopolymerization or photocuring process [84,85,86]. Photopolymerization is based on the transition of liquid-phase ink to solid-phase under UV or visible light irradiation. In general, oligomers-based inks are used for photopolymerization such as polyurethane, polyether, polyester, and epoxy resin [87]. Yu Zhang et al. reported one droplet UV-assisted printing by using bottom-up UV-illuminated printing. Figure 8a depicts the typical experimental setup of the one-droplet UV-illuminated printing using a UV-curable resin droplet. It is developed on three different substrates such as F-quartz, superamphiphobic, and S-PDMS as shown in Figure 8b–s. The authors have found that one droplet UV illuminated resin grown on F-quartz and superamphiphobic surface has ruptured curing and curing with vertical striped patterns, the UV curing on S-PDMS substrate has better resolution than the other counterpart (Figure 8b–s). In another study, Junzhe Zhu et al. reported the NIR laser (980 nm) assisted photo polymerization technology for the development of small feature sizes of 4 mm 3D complex structures as shown in Figure 8t. The authors showcased the scaling ability of the NIR laser by developing a mesh-like pattern with a line width of ~400 µm as depicted in Figure 8u,v. Similarly, the honeycomb-like structure is also showcased in 3D-printed monolithic structures at different resolutions as shown in Figure 8w. Figure 8x,y shows the freestanding cantilever structure with color different pigments, the proposed structure represents the free-standing “m” with direct ink writing technology.

## 4. Printing Quality

The printing quality of the lines is the crucial factor that determines whether the printing technology is reliable or not [90]. However, the quality of the printed metal line can be categorized into three classifications such as fine line, uneven edges, and overspray as shown in Figure 9 [90]. Fine lines are perfectly edged metal lines without any deformations or ruptures on the side wall of the metal lines. On the other hand, uneven edges are exhibiting curvy features with uneven width from one region of the metal line to another. An overspray scenario generally happens when the line width exceeds the desired line width. 

## 5. Coffee Ring Effect

Inkjet and aerosol printers are suffering seriously from the coffee ring effect as the name suggests the nanoparticle printing ink on reaching the surface forms a coffee ring-like circular pattern of nanoparticle ink on the substrate [91,92]. In general, the coffee ring effect is mostly observed when using an inkjet printer and an aerosol jet printer. Rafal Sliz et al. predicted the coffee ring effect by varying the substrate temperature from 25 °C to 250 °C as shown in Figure 10a–f [93]. The authors used the PEDOT: PSS as ink materials and their findings suggest that the coffee ring effect is larger when the substrate temperature is at 130 °C. Conversely, the coffee ring effect is suppressed when the substrate temperature is >130 °C which is due to the decrease in heat transfer between the substrate and the droplet. The coffee ring effect reappears when the temperature is greater than the second critical point of 220 °C. The coffee ring effect affects the device performance by a non-uniform coating of materials leading to the poor electrical performance of the devices.

As reported by Pei He and Brian Derby, the coffee ring effect mitigation on modulating the drying temperature and mean GO flake size is depicted in Figure 10g. It shows no coffee ring effect on increasing the drying temperature and mean GO flake size. The coffee ring effect can be considerably reduced by choosing a larger flake or nanoparticle size and also by modulating substrate treatment temperature. Tony M. Yen et al. [95] reported the coffee ring effect reversal by using CO_2_ laser-induced differential evaporation. Moreover, the modulation of surface tension for PEDOT: PSS reduces the coffee ring effect [96]. Guo Liang Goh et al. [66] reported the SWCNT-based conducting ink used as the conducting single-wall carbon nanotube line by aerosol jet printing as shown in Figure 11a. The side view of the coffee ring effect on the patterned line with a thicker side wall and thinner center is observed in Figure 11b,c. In this study, the patterned lines have an uneven side wall with ruptures and deformations at the edges leading to the formation of unstable lines this is predominantly due to the coffee ring effect as shown in Figure 11d–h. The sheath flow of the aerosol jet printing determines the resistance of the SWNT. The coffee ring effect width can be controlled by modulating the printing speed, nozzle size, and substrate temperature as shown in Figure 11i–j.

## 6. Applications of 3D-Printing Technology

3D-printing technology is widely adopted in various industries from construction to prototyping. Similarly, 3D printing is going to achieve momentum in the manufacturing of electronic products for various applications. In addition, conducting ink is gaining rapid momentum around the globe in recent years due to the rise in requirements across various application domains based on flexible and Si-based electronics. The emergence of printable electronics on flexible substrates opens a wide range of applications such as biomedical devices, biosensors, thin film transistors, solar cells, sensors, actuators, photodetectors, energy devices such as supercapacitors, and fuel cells as shown in Figure 12. Even aerosol jet printing is widely used in solar cell metallization applications [97]. Moreover, aerosol jet printing is highly suitable for semiconductor packaging, interconnects, and metallization [19,97,98]. Thus, by introducing printed metal interconnects and metallization into the commercial fabrication processing, we are reducing the hard-to-recyclable toxic metal and chemical waste which are produced during large-scale lift-off, etching, and metallization process. It also improves efficiency by minimizing the operating cost and chemical costs during the back end of the line process which further decreases the overall production cost of a chip on a larger scale. Printing technology is recently adapted for the fabrication of parasitic electronics and energy storage devices. In a study reported by Sung-Yueh Wu et al., the author demonstrated the 3D-printable passive electronics such as resistors, capacitors, and inductors for wireless integrated sensors [99]. Similarly, incorporating printing technology into discrete electronic will reduce the barrier between creators and their creations. This will create an open innovation platform for younger innovators to develop next-generation products for various application sectors.

Printed fuel cells based on protonic ceramics are attracting interest in clean energy and a sustainable future [100,101,102]. Furthermore, compact and printable batteries and fuel cells further provide pathways for next-generation innovators to develop cutting-edge applications across various nodes. The technology growth allows the sensor to play a prominent role in the day-to-day activities of human; printed microsensors for sensing temperature [103], strain [104], and photonic sensors [105,106,107]. For various biomedical applications, printed sensors are highly desirable for simple and rapid manufacturing processes [108,109,110].

## 7. Conclusions

Printed electronics are likely to revolutionize electronic manufacturing and prototyping by acting as a bridge between laboratory-scale innovation and real-time application. AM technology is a highly potential technology for the fabrication of next-generation electronics using low cost and rapid prototyping ability for cross-functionality oriented applications. The printed electronics predominantly relate to those specific requirements of the printing process and can substantially utilize any material, that can be deposited by solution-based, air-based, or other processes which are comprehensively assessed in the manuscript. The key highlights are as follows:Systematic assessment in the past 10 years based on emergent printed electronics from the perspective of materials and various processing techniques. In addition, various ink materials with superior electrical and mechanical properties to ensure their pathways in the electronics industry are discussed.The prominent role of printing ink and its classification based on the ink materials are also discussed. We conducted a comprehensive review of metal oxide nanoparticle ink, metal nanoparticle ink, 2D material ink, molecular ink, and UV-curable ink. Among them, metal nanoparticle inks are highly desirable for high-conducting metal lines and electrodes. Conversely, metal oxide nanoparticle ink can exhibit conducting and dielectric nature. Molecular and UV-curable ink has a flexible nature which makes it highly suitable for flexible electronics applications.A critical assessment on the potential of diverse printing technology and its ability in contributing to the semiconductor industry for sustainable manufacturing are presented. In addition, we discussed the role of printed metal lines and electrodes by inkjet, aerosol, and electrohydrodynamic printing for the state-of the art semiconductor process and interconnects. We discussed the correlation of the surface tension of the printing ink for nanometer-scaled resolution using electrohydrodynamic printing.The printing quality of the metal line and their classification such as fine lines, uneven edges, and overflow are also holistically discussed. Also, the prominent role of the coffee ring effect in inkjet and aerosol jet printing technology and also various methods proposed to mitigate the effect are envisaged.The present work will be helpful in providing insight into this emerging technology to the spectrum of researchers and industrial professionals. The highly stable and reliable features of printed conductive inks imply a promising future in flexible electronics and their applications. We also discussed the various applications of printed electronics in electronic devices, energy devices, sensors, and biomedical devices which have promising applications.

Finally, we conclude that printed electronics is a highly efficient and sustainable alternative technology for industry 5.0. AM technology which will provide sustainable growth to electronic industries by minimizing global chemical waste across electronic manufacturing sectors. In the future, the printable electronic will be revolutionizing the world by providing freedom to innovate even for mass production for an open innovator ecosystem.

## Figures and Tables

**Figure 1 nanomaterials-12-04251-f001:**
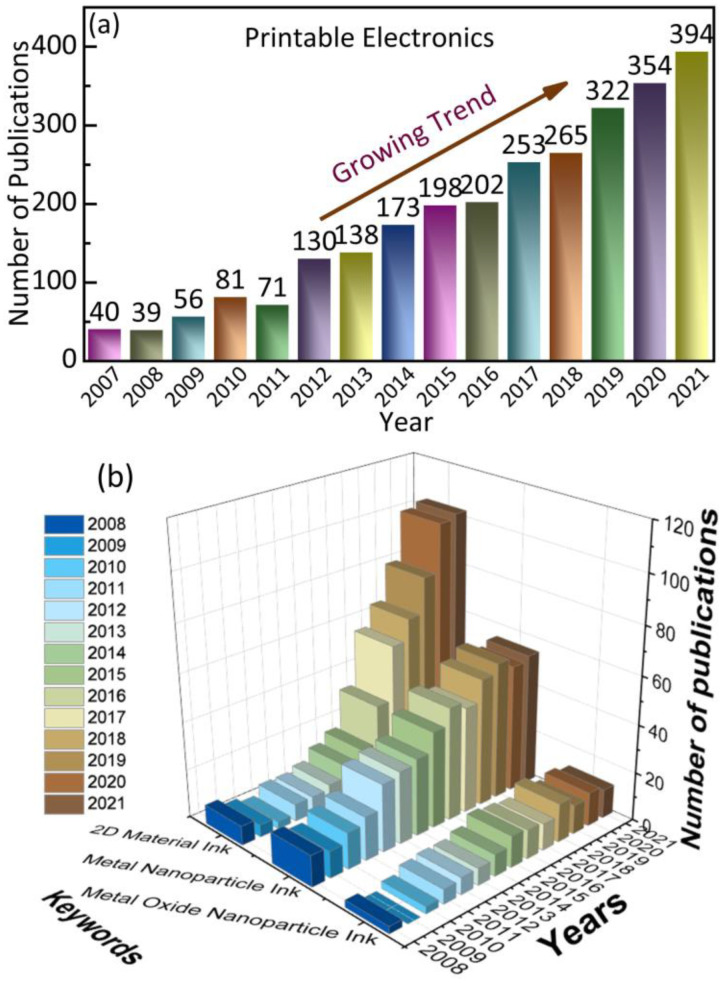
Recent publications in the Web of Science database using keywords, (**a**) printable electronics and (**b**) 2D material ink, metal nanoparticle ink, and metal oxide nanoparticle ink.

**Figure 2 nanomaterials-12-04251-f002:**
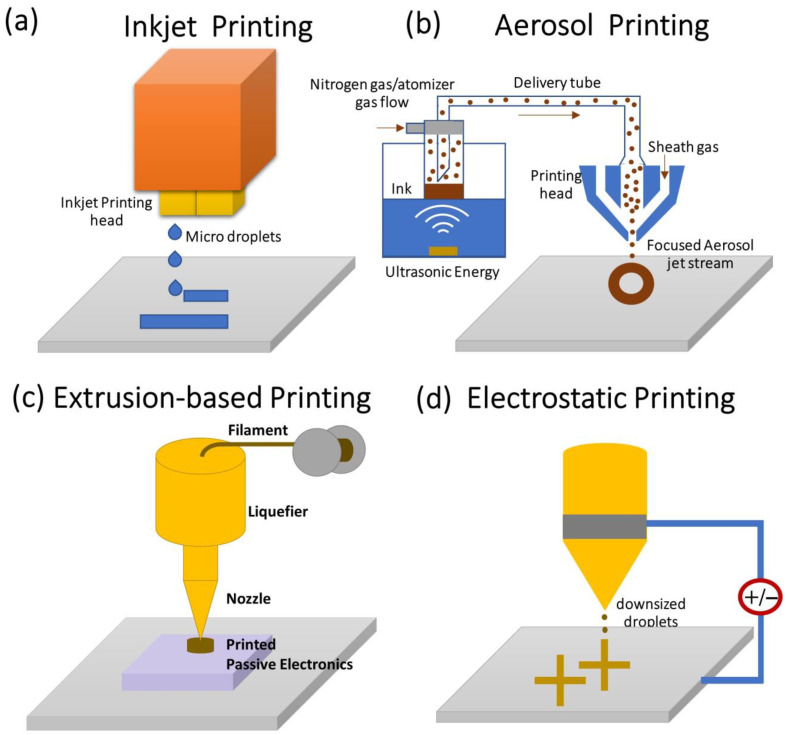
Printing technologies using conducting ink (**a**) inkjet printing; (**b**) aerosol printing; (**c**) ex-trusion-based printing; and (**d**) electrohydrodynamic printing. Adapted from [61].

**Figure 3 nanomaterials-12-04251-f003:**
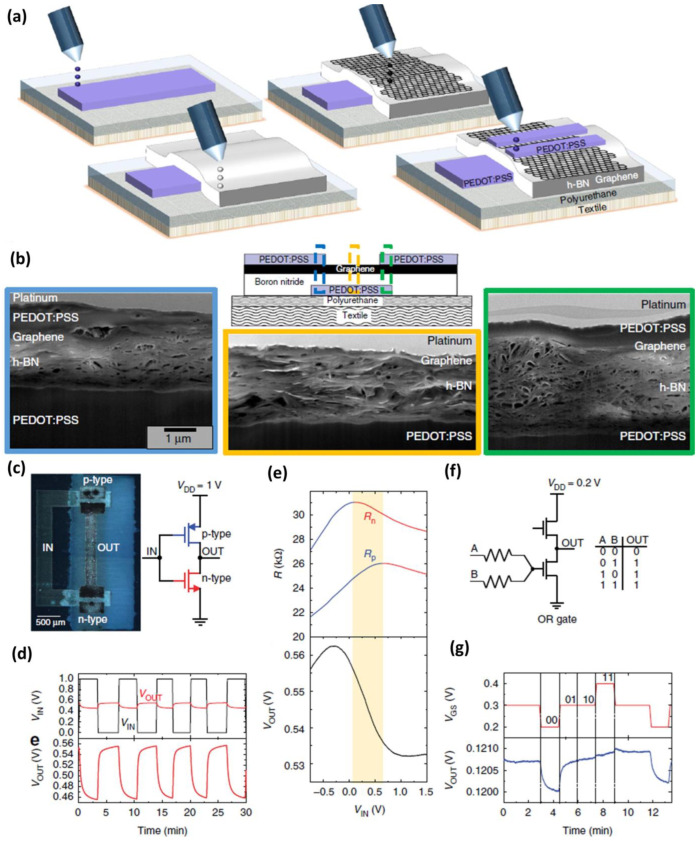
(**a**) Inkjet printer heterojunction transistor with graphene printed by inkjet printing technology; (**b**) cross-sectional SEM image of FET fabricated on the textile substrate; (**c**,**d**) optical microscopic image of the inverter and inverter input/output waveform based on FET; (**e**) measure p and n-type resistance on the applied input voltages; (**f**) OR logic gate diagram and truth table: where A & B are the input of the OR gate and OUT as the output logic with logic 0 as OFF and logic 1 as ON; (**g**) OR logic gate digital logical transition waveform (reprinted with permission, Copyright 2017 Springer Nature [62]).

**Figure 4 nanomaterials-12-04251-f004:**
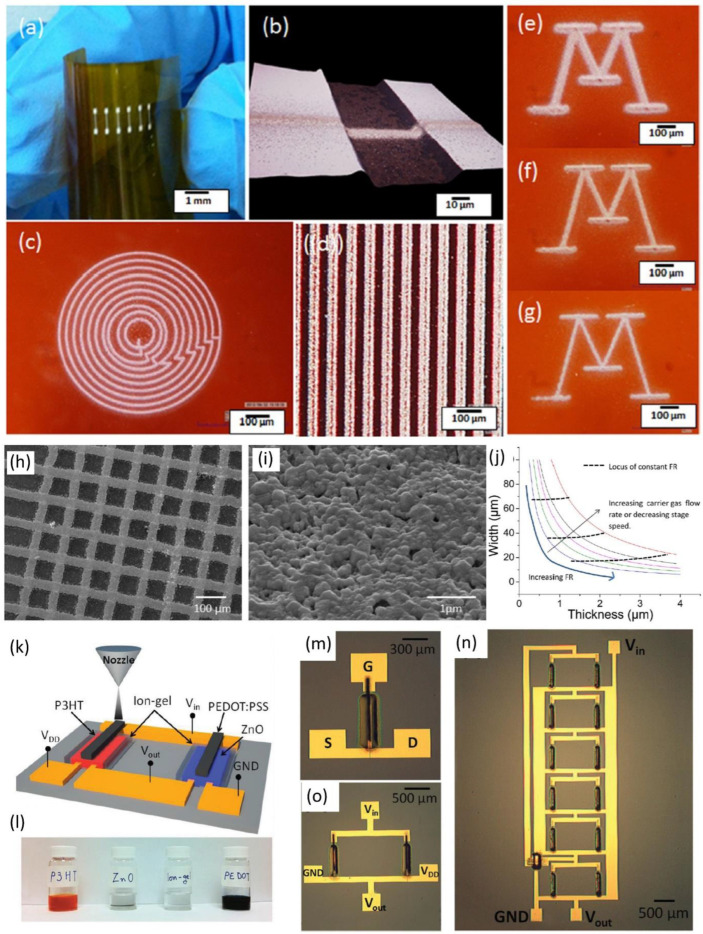
(**a**–**g**) Aerosol jet printed silver metal line and patterns on the polyimide substrate; (**h**,**i**) SEM image of the metal line; (**j**) relationship of focus ratio on thickness and width of the metal line; (reprinted with permission, Copyright 2013 American Chemical Society [67]); (**k**) aerosol jet printed ZnO based transistor; (**l**) inks used for the aerosol jet printing; (**m**–**o**) optical microscopic image; (reprinted with permission, Copyright 2014 John Wiley and Sons [64]).

**Figure 5 nanomaterials-12-04251-f005:**
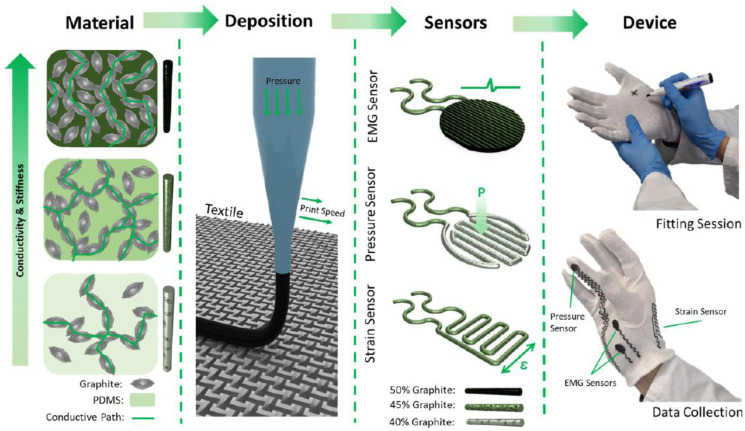
Fused deposition of graphene-based flexible conducting lines on wearable gloves as sensors; (reprinted with permission, Copyright 2022 IOP Science [49]).

**Figure 6 nanomaterials-12-04251-f006:**
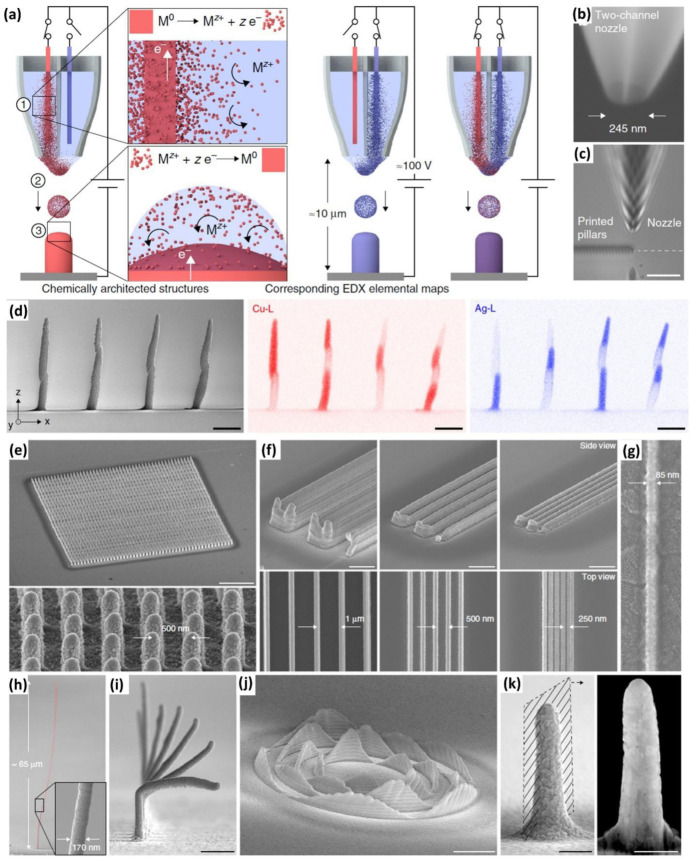
(**a**) Working principle of electrohydrodynamic redox printing with two-channel operating mode: (1) Metal ions generated and immersed in liquid solvent, (2) ionized solvents are ejected as droplets by electrohydrodynamic force, (3) On substrates the metal ions are reduced to form zero valence metal; (**b**,**c**) optical microscopic image of the two-channel nozzle; (**d**) SEM micrograph and EDX elemental mapping of nanopillar based on Cu and Ag; (**e**) 50 × 50 copper pillars array; (**f**) 250 nm spaced wall printed; (**g**) Cu metal lines of 100 nm; (**h**,**i**) overhangs pillar having out of plane growth; (**j**) sine wave pattern printed on the surface; and (**k**) cu nanopillar with polycrystalline microstructure; (reprinted with permission, Copyright 2019 Springer Nature [69]).

**Figure 7 nanomaterials-12-04251-f007:**
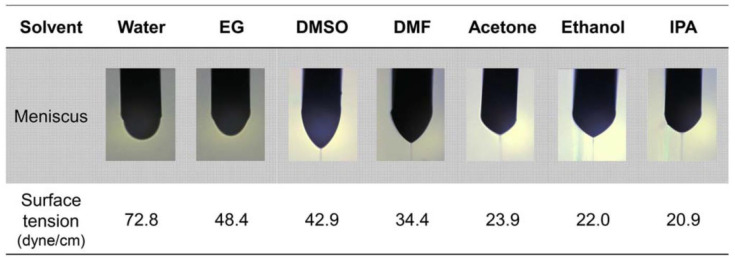
Printable ink with various surface tension effects on electrohydrodynamic printing head jetting formation (reprinted with permission, Copyright 2017 John Wiley and Sons [83]).

**Figure 8 nanomaterials-12-04251-f008:**
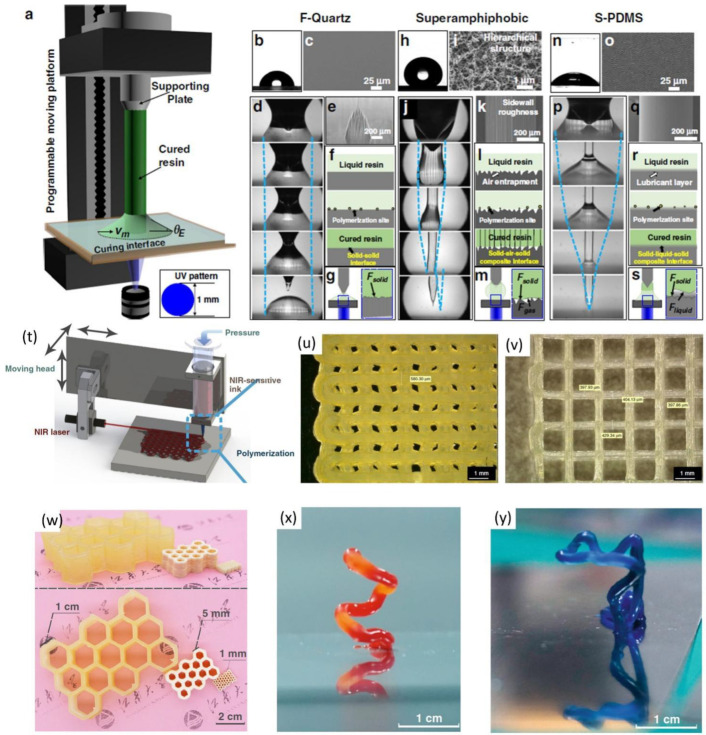
(**a**) Single droplet UV-based printing system; (**b**–**s**) bottom-up printing approach on F-quartz, superamphiphobic, and S-PDMS substrate; (reprinted with permission, Copyright 2020 Springer Nature [88]); (**t**) NIR based direct ink writing technology; (**u**,**v**) post-NIR exposure to form mesh-like structure; (**w**) 3D monolithic honeycomb structure side and top view; (**x**,**y**) freestanding spiral and m shaped cantilever structure with red and blue pigment. reprinted from (reprinted with permission, Copyright 2020 Springer Nature [89]).

**Figure 9 nanomaterials-12-04251-f009:**
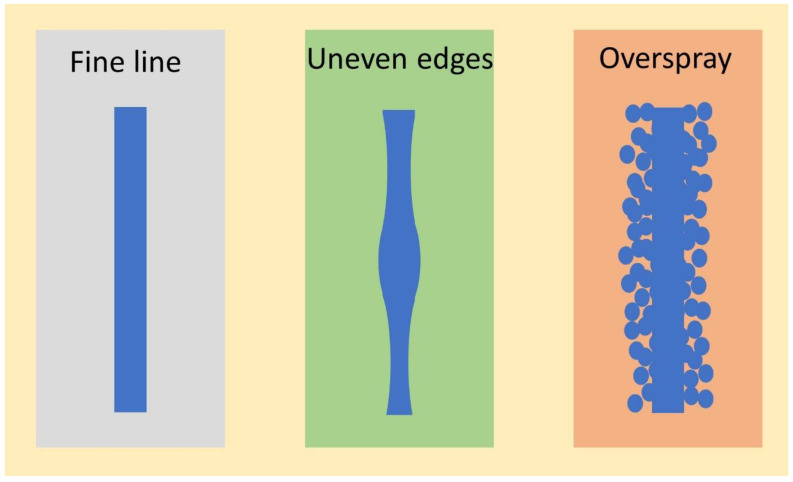
Printed patterns quality such as fine lines, uneven edges, and overspray.

**Figure 10 nanomaterials-12-04251-f010:**
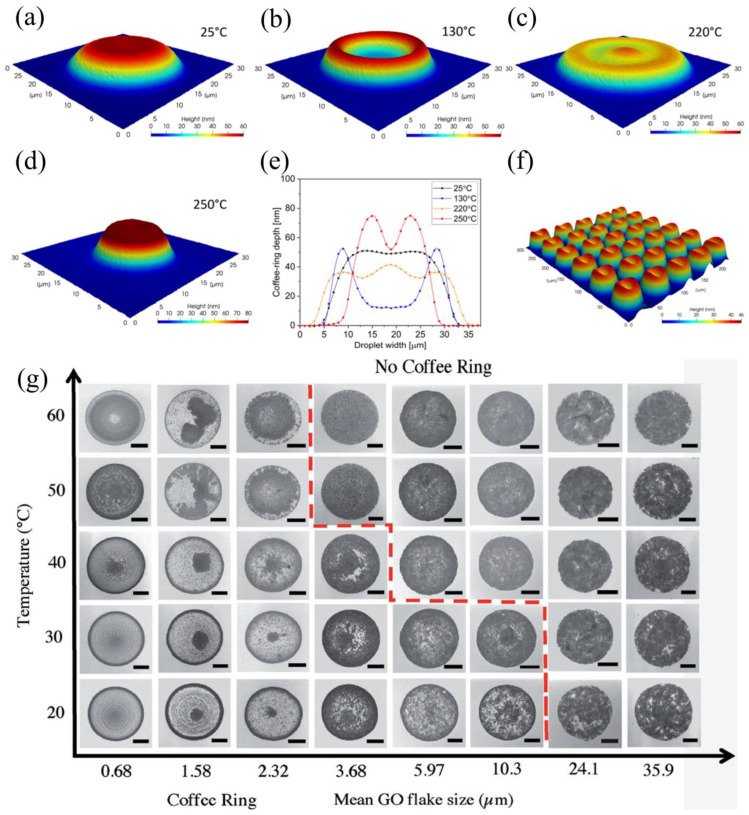
(**a**–**f**) Coffee ring effect associated with the heating temperature on inkjet-printed microdroplets; (reprinted with permission, Copyright 2020 American Chemical Society [93]); (**g**) coffee ring effect mitigation by using the thermal treatment on GO flake size; (reprinted with permission, Copyright 2017 John Wiley and Sons [94]).

**Figure 11 nanomaterials-12-04251-f011:**
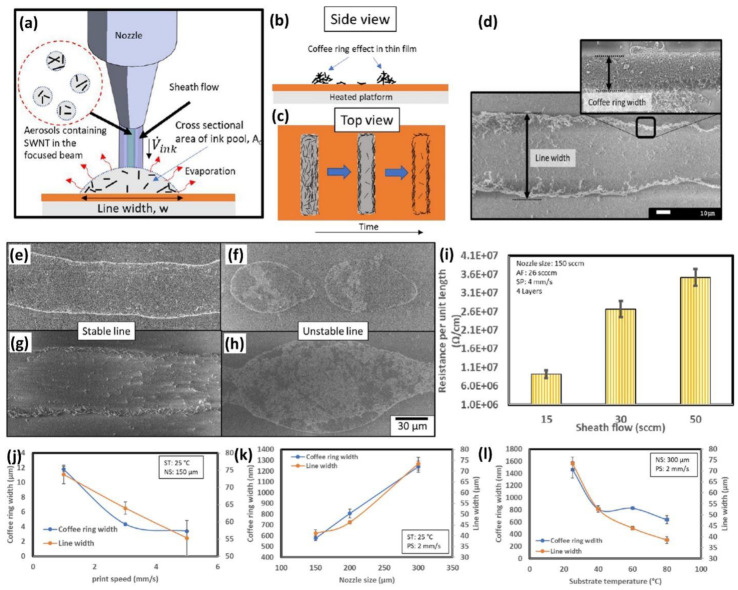
(**a**) Aerosol jet printing process schematic using SWNT; (**b**,**c**) Side view and top view of coffee ring effect; (**d**–**h**) SEM top view of stable and unstable line with coffee ring width; (**i**) dependence of sheath flow and the resistance per unit length of the printed line; (**j**) coffee ring width and print speed; (**k**) coffee ring width and nozzle size; (**l**) coffee ring width and substrate temperature; (reprinted with permission, Copyright 2019 American Chemical Society [66]).

**Figure 12 nanomaterials-12-04251-f012:**
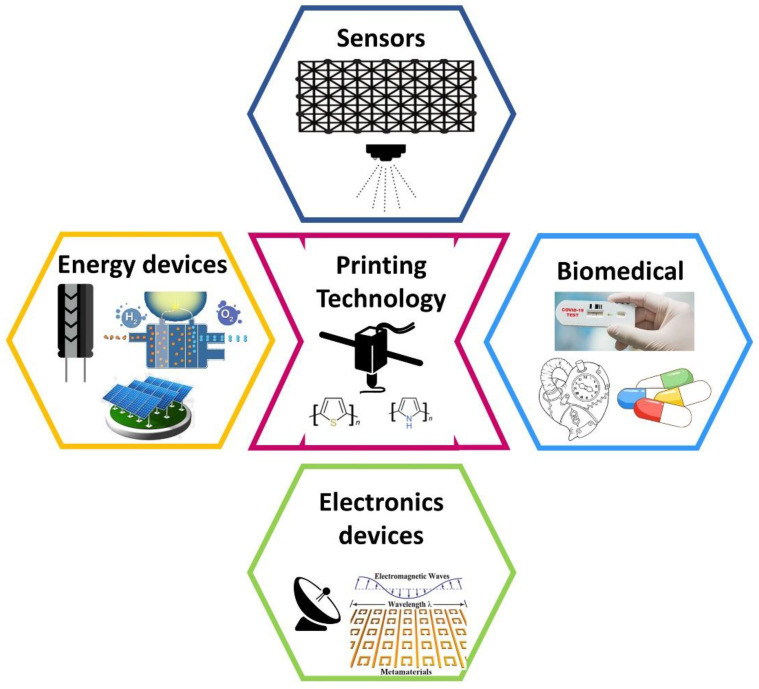
Applications of printed electronics as sensors, biomedical devices, energy devices, and other communication electronics such as antennas.

**Table 1 nanomaterials-12-04251-t001:** Comparison of nanoparticle ink used in various printing technologies.

Authors	Ink Materials	Properties	Technique	Nozzle Size
Tian Carey et al. [62]	h-BN, and graphene ink	Viscosity _h-BN_—1.7 mPa s, Viscosity _graphene_—1 mPa s,	Inkjet printing	21 μm diameter nozzle
Samia Mekhmouken et al. [20]	Gold nanoparticle ink	Viscosity—2.75 mPa s Surface tension—34.2 mNm^−1^	Inkjet printing	NA
Iara J. Fernandes et al. [21]	Silver nanoparticle ink	η—3.7 to 7.4 mPas Surface tension—33 to 36 mNm^−1^	Inkjet printing	NA
Jin Sung Kang et al. [63]	Copper nanoparticle ink	Resistance—36.7 nΩ m	Inkjet printing	NA
Subimal Majee et al. [27]	Zinc nanoparticle ink	Surface tension—25 mNm^−1^ Viscosity—5 mPa s	Inkjet printing	21 μm diameter nozzle
Guohua Hu et al. [12]	Black phosphorus ink	Surface tension—28 mNm^−1^ Viscosity—2.2 mPa s	Inkjet printing	22 μm diameter nozzle
Kihyon Hong et al. [64]	Zinc oxide nanoparticle ink	Resistance—25 Ω cm	Aerosol Jet Printing	250 µm diameter
Xuewei Feng et al. [65]	MoS_2_, and Ag nanoparticle ink	NA	Aerosol-Jet Printing	300 µm diameter
Guo Liang Goh et al. [66]	Carbon nanotube ink	Resistance—600 kΩ/cm	Aerosol-Jet Printing	150 µm diameter
Ankit Mahajan et al. [67]	Silver nanoparticle ink	Resistance—3.61 μΩ cm	Aerosol-Jet Printing	100, 150, and 200 µm diameter
Li Tu et al. [68]	Silver nanowire ink	Sheet resistance—57.68 Ω/sq	Aerosol-Jet Printing	200 µm diameter
Pavel V. Arsenov et al. [26]	Platinum ink	Surface tension—44 mNm^−1^ Viscosity 11 cP Resistivity—1.2 × 10^−7^ Ωm	Aerosol-Jet Printing	300 µm diameter
Alain Reiser et al. [69]	Cu, and Ag	NA	Electrohydrodynamic printing	245 nm diameter
Thi Thu Thuy Can et al. [70]	Cu	Viscosity 4000 cPs Resistivity—8 × 10^−4^ Ωm	Electrohydrodynamic printing	100 µm diameter
Sangkyu Lee et al. [71]	IZO	NA	Electrohydrodynamic printing	2 µm diameter

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
