# Peer review of "A Comprehensive Review on Printed Electronics: A Technology Drift towards a Sustainable Future"

_nanomaterials, 2022, doi:10.3390/nano12234251_

Round 1

Reviewer 1 Report

Dear authors,

Thank you for your submission. I hope that my remarks will help you improving the quality of the paper.

1. The title should be rephrased and specified in it that this is a narrative review.

2. The abstract should be extended.

3. Why are two corresponding authors and none of them the last author? Usually the last author is the corresponding one.

4. Abstract should be improved.

5. Line 38. Two many references for such a short and simple statement. Please elaborate.

6. Please indicate how did you select the articles used for this narrative review.

7. From my point of view figure 1 should be redesigned in a much elegant and academic style.

8. Is figure 2 original? If so the pieces of information don't require citations?

9. Please provide the reprinting acceptance for figures 3-11.

10. Rephrase the Conclusions section and remove Outlook. Please try to emphasise the perspectives that this review opens and the novelty of it.

Author Response

We have submitted our point-by-point response to the reviewer's comments in the enclosed attachment. Also, attached are copyrights of reprinting permission of each article from page 4 to 27 in a sequential manner. 

Thank you

Reviewer 2 Report

The work is very interesting and meets the requirements of the journal Nanomaterials. 

Author Response

Thank you for your kind and valuable comments. Please kindly check the enclosed attachment for the revised version of the manuscript. 

Reviewer 3 Report

To improve the quality of the manuscript, it would be a good thing to introduce the following revisions:

Please expand the “Abstract” section and describe in more detail what was done and shown in this review.

All the abbreviations, used in the manuscript, must be interpreted or a glossary must be provided in the manuscript.

At the very beginning of the manuscript, on the first page, in the “Introduction”, you are writing straight away about the advantages and prospects of additive technologies. But isn't it as a result of the review, comparing the pros and cons of various technologies, that you should arrive at this conclusion? It would be more appropriate to indicate the need for such review and pose a problem in the “Introduction” section.

In lines 70-80, you give quite a lot of statements. It would be better to confirm these statements by references.

In lines 83-84 you are writing about the fact that inks with nanoparticles are preferable. But it is more appropriate to write about the pros and cons of that. Or you may indicate in which cases and for which application they are preferable. The exact indication of the areas of application in such cases should be followed throughout the article.

For example, in lines 133-135 you are writing that the works [44] and [45] contain information about inks on different bases. But what is contained in these studies and what kind of contribution they made  are not given in the text. In the review, it would be a good thing to give at least a little information about the achieved results and problems solved by the researchers. And there is no need to list who and on what topic carried out the work. It is worth correcting throughout the entire the manuscript.

There are no conclusions for each section. In the review, it is necessary to introduce something new based on the presented material: comparative analysis, development prospects, etc. Conclusions on the work are very general and they do not follow from the research material provided in the manuscript. It would be right to correct the conclusions. Please, add that based on such-and-such things, presented in the review, we conclude about …

Author Response

(The authors gave the same response as above.)

Round 2

Reviewer 1 Report

Dear authors,

Thank you for providing the revised version of the manuscript.

My piece of advice, in order to increase the quality and the impact of the paper is to kindly use the MDPI English Editing service.

Best of luck with your work!

Author Response

We would like to express sincere thanks to all the reviewers for the comments and suggestions for this present assessment. The modifications made pointwise are listed below.

Reviewer 1

Thank you for providing the revised version of the manuscript.

My piece of advice, in order to increase the quality and the impact of the paper is to kindly use the MDPI English Editing service.

Best of luck with your work!

Thank you for your valuable comments.

As suggested by the reviewer, the English standard of the manuscript has been edited by an English professional.

Reviewer 3 Report

Thanks to the authors for the good work on the review, but I ask you to make a few more improvements:
1. Slightly expand sections 2.3 and 2.5. Write in more detail what limits the distribution of these inks, and what are their disadvantages.
2. You have added a description on lines 141-157 (refs 44 and 45). Similarly, it is worth doing for lines 252-255 (references 71, 72). On lines 410-419, you list the possibilities for using the described technologies. It would also be nice to expand the text a bit, indicating the advantages of such widespread use.
3. The text on lines 344-351 is better to confirm with a link. Similarly, 356-360.
4. The conclusions of my remark have not been practically finalized:
There are no conclusions for each section. In the review, it is necessary to introduce something new based on the presented material: comparative analysis, development prospects, etc. The conclusions of the work are very general, and they do not follow the research material provided in the manuscript. It would be right to correct the conclusions. Please, add that based on such-and-such things, presented in the review, we conclude about...
The conclusions should be expanded and a conclusion should be added at the end of each large section summarizing the work done (reflecting the prospects for developing technologies, possible new areas of application, etc.).

Author Response

We would like to express sincere thanks to all the reviewers for the comments and suggestions for this present assessment. The modifications made pointwise are listed below.

Round 3

Reviewer 3 Report

Thanks to the authors for their work. The authors corrected my comments.